# Contrasting factors associated with COVID-19-related ICU admission and death outcomes in hospitalised patients by means of Shapley values

**Massimo Cavallaro**[1,2]*, **Haseeb Moiz**[3], **Matt J. Keeling**[1,2], **Noel D. McCarthy**[1,3,4]*

**1** The Zeeman Institute for Systems Biology & Infectious Disease Epidemiology Research, University of Warwick, Coventry, United Kingdom, **2** School of Life Sciences and Mathematics Institute, University of Warwick, Coventry, United Kingdom, **3** Warwick Medical School, University of Warwick, Coventry, United Kingdom, **4** Institute of Population Health, Trinity College Dublin, University of Dublin, Dublin, Ireland

* M.Cavallaro@warwick.ac.uk (MC); Noel.Mccarthy@tcd.ie (NDM)

## Abstract

Identification of those at greatest risk of death due to the substantial threat of COVID-19 can benefit from novel approaches to epidemiology that leverage large datasets and complex machine-learning models, provide data-driven intelligence, and guide decisions such as intensive-care unit admission (ICUA). The objective of this study is two-fold, one substantive and one methodological: substantively to evaluate the association of demographic and health records with two related, yet different, outcomes of severe COVID-19 (viz., death and ICUA); methodologically to compare interpretations based on logistic regression and on gradient-boosted decision tree (GBDT) predictions interpreted by means of the Shapley impacts of covariates. Very different association of some factors, e.g., obesity and chronic respiratory diseases, with death and ICUA may guide review of practice. Shapley explanation of GBDTs identified varying effects of some factors among patients, thus emphasising the importance of individual patient assessment. The results of this study are also relevant for the evaluation of complex automated clinical decision systems, which should optimise prediction scores whilst remaining interpretable to clinicians and mitigating potential biases.

**Data Availability Statement:** Data on cases were obtained from the COVID-19 Hospitalisation in England Surveillance System (CHESS) data set that

## Author summary

The design is a retrospective cohort study of 13954 in-patients of ages ranging from 1 to 105 year (IQR: 56, 70, 81) with a confirmed diagnosis of COVID-19 by 28th June 2020. This study used multivariable logistic regression to generate odd ratios (ORs) multiply adjusted for 37 covariates (comorbidities, demographic, and others) selected on the basis of clinical interest and prior findings. Results were supplemented by gradient-boosted decision tree (GBDT) classification to generate Shapley values in order to evaluate the impact of the covariates on model output for all patients. Factors are differentially associated with death and ICUA and among patients. Deaths due to COVID-19 were associated with immunosuppression due to disease (OR 1.39, 95% CI 1.10–1.76), type-2 diabetes

collects detailed data on patients infected with COVID-19. These data contain confidential information, with public data deposition non-permissible for socioeconomic reasons. The CHESS data resides with the National Health Service (www.nhs.gov.uk). Interested readers wanting similar data may wish to contact coronavirus-tracker@phe.gov.uk. All codes for data management and analysis are archived online at https://github.com/mcavallaro/CovidC.

**Funding:** This work was supported by Health Data Research UK, which is funded by the UK Medical Research Council, EPSRC, Economic and Social Research Council, Department of Health and Social Care (England), Chief Scientist Office of the Scottish Government Health and Social Care Directorates, Health and Social Care Research and Development Division (Welsh Government), Public Health Agency (Northern Ireland), British Heart Foundation and the Wellcome Trust (MC, MJK, and NDM). MJK and NDM are affiliated to the National Institute for Health Research Health Protection Research Units (NIHR HPRUs) in Gastrointestinal Infections and in Genomics and Enabling Data. MJK is funded by UK Research and Innovation through the JUNIPER modelling consortium (MR/V038613/1). The views expressed are those of the author(s) and not necessarily those of the NIHR, the Department of Health and Social Care or Public Health England. The funders had no role in study design, data collection and analysis, decision to publish, or preparation of the manuscript.

**Competing interests:** The authors have declared that no competing interests exist.

(OR 1.31, 95% CI 1.17–1.46), chronic respiratory disease (OR 1.19, 95% CI 1.05–1.35), age (OR 1.56/10-year increment, 95% CI 1.51–1.61), and male sex (OR 1.54, 95% CI 1.42–1.68). Associations of ICUA with some factors differed in direction (e.g., age, chronic respiratory disease). Self-reported ethnicities were strongly but variably associated with both outcomes. GBDTs had similar performance (ROC-AUC, ICUA 0.83, death 0.68 for GBDT; 0.80 and 0.68 for logistic regression). We derived importance scores based on Shapley values which were consistent with the ORs, despite the underlying machine-learning model being intrinsically different to the logistic regression. Chronic heart disease, hypertension, other comorbidities, and some ethnicities had Shapley impacts on death ranging from positive to negative among different patients, although consistently associated with ICUA for all. Immunosuppressive disease, type-2 diabetes, and chronic liver and respiratory diseases had positive impacts on death with either positive or negative on ICUA. We highlight the complexity of informing clinical practice and public-health interventions. We recommend that clinical support systems should not only predict patients at risk, but also yield interpretable outputs for validation by domain experts.

## Introduction

COVID-19, due to SARS-CoV-2 betacoronavirus, emerged in Wuhan, China in late 2019 and has spread globally. It can cause severe complications of pneumonia, acute respiratory distress syndrome, sepsis, and septic shock [1]. It has, as of October 24, 2020, infected over 42 million people and killed over 1.1 million people [2]. Certain patient subsets, such as the elderly and those with comorbidities, are at an increased risk of severe outcomes from COVID-19 such as admission to intensive care units, respiratory distress requiring mechanical ventilation, and death [3,4].

Clinicians can use predictive factors to prioritize patients at higher risk of clinical deterioration and public health authorities can use them to target public health interventions. Identifying factors associated with severe disease has been described as an urgent research priority. Several studies have sought to identify factors predicting poor outcome following COVID-19 infection [5,6] and assist clinician decision making [7–9]. A traditional method such as logistic regression can infer the odd ratios (ORs) of the outcome in the presence of a risk factor. Modern machine-learning technologies, widely implemented during the COVID-19 pandemic, can handle more complex patient data types, offer greater generality, and produce more accurate predictions than the previous methods, but at the cost of losing transparency and interpretability [10].

Surveillance systems support these analyses. The COVID-19 Hospitalization in England Surveillance System (CHESS), a UK system distributed by Public Health England (PHE) and adapted from the UK severe influenza surveillance system, collects extensive data on patients admitted to hospital, including known comorbidities and important demographic information (such as age, sex, and ethnicity) [11]. This large national dataset reduces limitations inherent in small cohorts, enabling more reliable identification of associations. We performed analyses on this dataset using logistic regression and a more general machine-learning model (the gradient-boosted decision tree, GBDT), which generated interpretable predictions by means of the Shapley additive explanation, a technique that mitigates the interpretability issue in machine-learning outputs. For different applications of this technique to COVID-19 research see, e.g., references [12,13]. Through these methods, we demonstrated the extent to which pre-existing conditions differentially predicted death and intensive care unit (ICU) admission.

Some factors affected both similarly but others proved to be protective for one while increasing the risk for the other, or showed very different effect sizes. We also identified variation of effects among patients. These results may be useful to clinicians assessing hospitalized patients with COVID-19. They may also provide a greater context or benchmark for individuals evaluating or interpreting complex automated clinical decision systems designed to identify those most at-risk.

## Materials and methods

### Ethics statement

The data used in this study were supplied from the CHESS database after anonymisation under strict data protection protocols agreed between the University of Warwick and Public Health England. The ethics of the use of these data for these purposes was agreed by Public Health England with the Government's SPI-M(O) / SAGE committees.

### Description of cohort and outcomes

We studied a cohort of 13954 patients of which 8947 patients survived and 5007 died after contracting COVID-19. 5758 were admitted to ICUs, of whom 3483 were discharged after treatment, and 2275 died. The dataset includes epidemiological data (demographics, risk factors, and outcomes) on patients with a confirmed diagnosis of COVID-19 by 28th June 2020 who required hospitalization. We included all available chronic and pre-existing morbid conditions recorded by PHE as potential risk factors, including immunosuppression due to disease, asthma requiring medication, immunosuppression due to treatment, neurological conditions, respiratory conditions, obesity, type-1 and type-2 diabetes, hypertension, heart conditions, renal disease, liver diseases, and other comorbidities [11]. No acute illnesses or medical conditions were considered. In the CHESS dataset self-defined ethnicity is categorized according to the Office for National Statistics questionnaires into 17 factors, all included in the study. With 8628 patients, white British was the largest group in the cohort and therefore chosen as a reference category. 1895 patients did not identify themselves with any ethnicity and were labelled as "NA". With the exception of age and admission date, all features were stratified to binary variables. Entries labelled "diabetes" whose type was unknown and not recorded in the database as "type 1", have been considered as "type 2". Death and ICU admission were chosen as outcomes. The median age of this sample was 70 years (IQR 56–81, range 1–105), 59.25% were men and 0.18% had an unrecorded sex. The prevalence of comorbidities is reported in Table 1 and ethnicity in Table 2. Cross-correlations between recorded ethnicities and pre-existing conditions are illustrated in Fig 1.

### Statistical analysis

Logistic regression models were used to estimate odd ratios (ORs) of all 37 pre-existing conditions and demographic factors for both outcomes. Standard errors (SEs) and confidence intervals (CIs) of the ORs were computed using the Taylor series-based delta method and the profile likelihood method, respectively, and statistical significance assessed using the Benjamini-Hochberg (BH) test with false discovery rate set to 0.05 [14].

In addition, we applied a "gradient boosted decision tree" (GBDT) machine-learning model with logistic objective function, as an appropriate machine learning approach. A GBDT aggregates a large number of weak prediction models, in this case decision trees, into a robust prediction algorithm, where the presence of many trees mitigates the errors due to a single-tree prediction. Each individual tree consists of a series of nodes that represent binary decision

**Table 1. Fraction of patients in cohort by sex and comorbitidies.**

| | |
|---|---|
| Sex male | 0.593 |
| Other comorbidity | 0.315 |
| Hypertension | 0.270 |
| Chronic heart disease | 0.161 |
| T2 diabetes | 0.159 |
| Chronic respiratory disease | 0.109 |
| Obesity (clinical) | 0.106 |
| Chronic neurological cond. | 0.087 |
| Chronic renal disease | 0.084 |
| Asthma | 0.084 |
| Immunosuppression treatment | 0.030 |
| Immunosuppression disease | 0.027 |
| Asymptomatic testing | 0.021 |
| Chronic liver | 0.017 |
| T1 diabetes | 0.012 |
| Pregnancy | 0.006 |
| Serious mental illness | 0.006 |
| Sex unknown | 0.002 |

splits against one of the input variables, with its final output being determined by the nodes at the end of the tree (known as leaves). The model was implemented in the XGBoost library (version 0.81) [15] and depended on a number of hyper-parameters. To avoid over-fitting, these hyper-parameters were selected by means of Bayesian optimization of c-statistics using 5-fold cross-validation over the training set [16] with constant L1-regularisation parameter $\alpha$ = 0.5. We used Shapley additive explanation (SHAP) analysis to understand the result of a GBDT model fit [17,18]. The importance of each feature in the model output is represented by the so-called Shapley values, introduced in game theory literature and providing a theoretically

**Table 2. Fraction of patients in cohort by ethnicity.**

| | |
|---|---|
| White British | 0.598 |
| Eth. NA | 0.134 |
| Eth. unknown | 0.102 |
| Other white | 0.026 |
| Other Asian | 0.024 |
| Other ethn. | 0.024 |
| Indian | 0.024 |
| Pakistani | 0.019 |
| Black African | 0.013 |
| Black Caribbean | 0.010 |
| Other black | 0.006 |
| White Irish | 0.004 |
| Other mixed | 0.004 |
| Bangladeshi | 0.004 |
| White and black Caribbean | 0.003 |
| Chinese | 0.003 |
| White and black African | 0.002 |
| White and Asian | 0.002 |

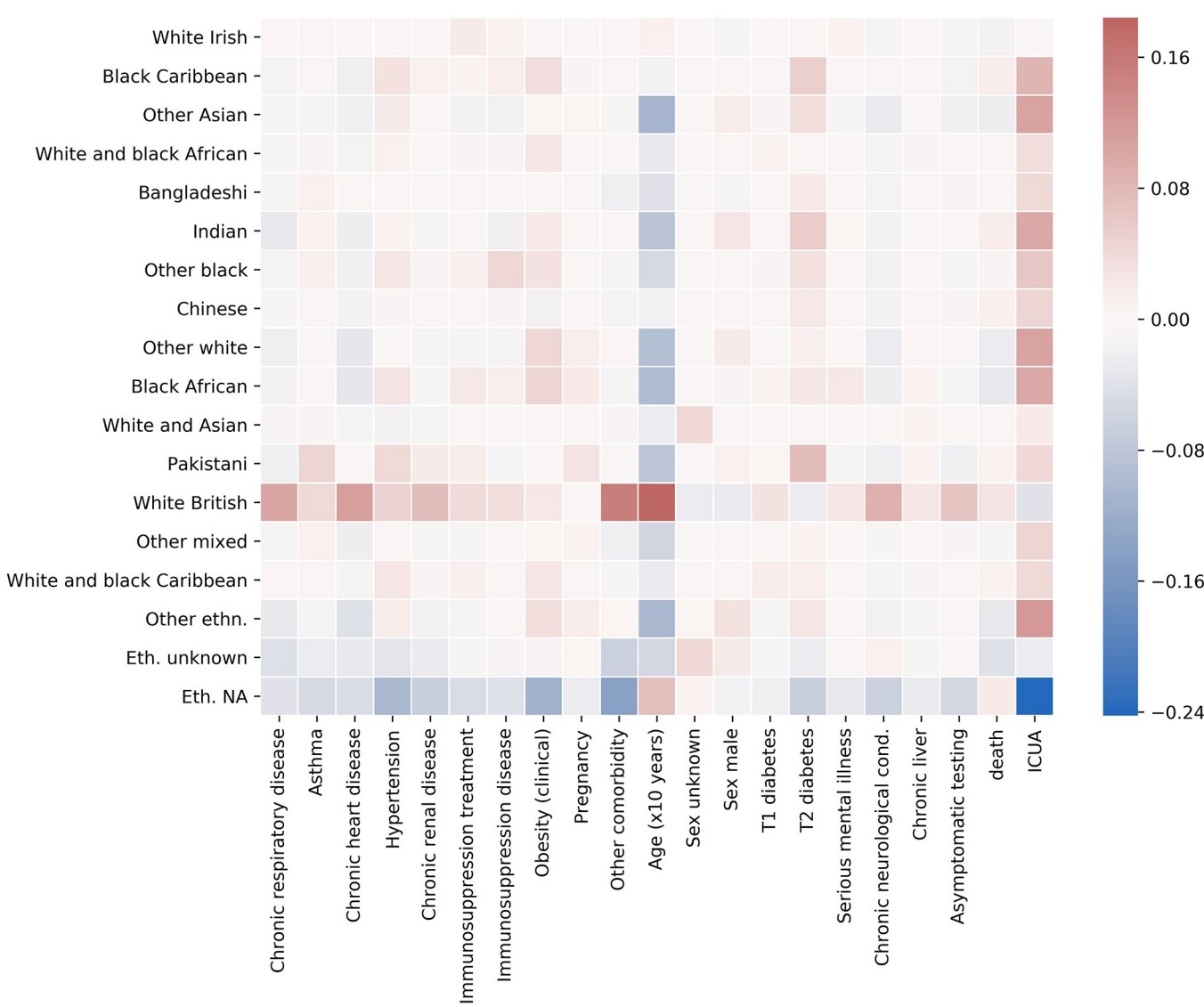

**Fig 1. Correlation heatmap between self-defined ethnicities and pre-existing conditions.** Colour shades from blue to red correspond to increasing values of Person correlation coefficient (white: no correlations are present). NA labels inpatients who did not identify themselves with any ethnicity.

justified method for allocation of credit among a group of players. In the context of machine learning, the same mathematics is used to allocate the credit for the GBDT prediction among the $N$ features included in the study, for each of the $M$ patients. The chief output of this approach is a $M \times N$ matrix of Shapley values $\phi_{ij}$ where $i$ indicates a patient, $i = 1,2,\ldots,N$, and $j$ is a pre-existing condition or other patient characteristic, $j = 1,2,\ldots,N$. We also refer to the Shapley value $\phi_{ij}$ as the impact of $j$ on the outcome for the patient $i$. Similar to the logistic regression model, for each patient $i$, the trained GBDT model returns a decision value $f_i$ to be interpreted as the logarithm of the odds that the outcome is poor. The Shapley values are unique allocations of credit in explaining the decision $f_i$ among all the $N$ features, where for our case, negative values ($\phi_{ij} < 0$) tip the decision value towards good outcome, while positive values ($\phi_{ij} > 0$) towards bad (i.e., ICU or death). The model output satisfies $f_i = \sum_{j=0}^{N} \phi_{ij}$ (which is the local accuracy property), where $\phi_{i0}$ is a bias term. Importantly, it has been

mathematically proven that the Shapley allocation is the only possible one that satisfies two additional desirable properties, i.e., consistency (if a feature's contribution increases or stays the same regardless of the other inputs, its Shapley value does not decrease), and missingness (a zero-valued feature contributes a zero Shapley value) [17–19]. In tree-based models, the same idea has been extended to allocate the credit to pairs of features, thus yielding $f_i = \sum_{k=0}^{N} \sum_{j=0}^{N} \Phi_{ijk}$, where the $\Phi_{ijk}$s are referred to as SHAP interaction values [18]. The diagonal term $\Phi_{ijj}$ encodes the net effect on the model prediction $f_i$ of a feature $j$, stripped of its interactions with the other features $k \neq j$ and is referred to as the SHAP main effect of $j$. We used an implementation specific to tree-based models, also referred to as TreeSHAP, accessible via the XGBoost and SHAP libraries; we refer the reader to references [17,18] for a more comprehensive discussion and for the implementation details.

Such an approach explains each individual prediction $f_i$ and is therefore referred to as a *local* method. In contrast to that, as a complementary *global* method, we consider the so-called partial dependence plots (PDPs) to show the average effects of age and admission date on the predicted outcomes, marginalizing over the values of all other features [20].

It is worth comparing this approach with the standard logistic regression. For a patient $i$ with feature values $\mathbf{X}_i := (x_{i1}, x_{i2}, \ldots, x_{iN})$, the logistic regression and the GBDT models predict an outcome (here taken to be ICUA or death) with probabilities $p(\mathbf{X}_i)$ and $\tilde{p}(\mathbf{X}_i)$, respectively. These satisfy

$$\log \frac{p(\mathbf{X}_i)}{1 - p(\mathbf{X}_i)} = \beta_0 + \beta_1 \cdot x_{i1} + \beta_2 \cdot x_{i2} + \ldots + \beta_N \cdot x_{iN}$$

and

$$\log \frac{\tilde{p}(\mathbf{X}_i)}{1 - \tilde{p}(\mathbf{X}_i)} =: f_i = \phi_{i0} + \phi_{i1} + \phi_{i2} + \cdots + \phi_{iN},$$

where the coefficients $\beta_j$s are maximum-likelihood estimates and the values $\phi_{ij}$s are obtained by means of the TreeSHAP algorithm. To rank the features by their overall importance, we estimate the slopes $\phi_j \mathbf{x}_j^T / (\mathbf{x}_j \mathbf{x}_j^T)$ for each $j$, where $\phi_j := (\phi_{1j}, \phi_{2j}, \ldots, \phi_{Nj})$ and $\mathbf{x}_j := (x_{1j}, x_{2j}, \ldots, x_{Nj})$, thus obtaining a novel feature score which we refer to as $Imp_j$ and can be directly compared to the coefficient $\beta_j$.

All models were fitted to a randomly chosen 90% of data entries, while the remaining entries were used for validation. Goodness-of-prediction was assessed by means of the c-statistics of the receiver operating characteristic curve (ROC-AUC) on the validation set, with bootstrapped 2.5%-97.5% confidence intervals.

## Results

Risk factors showed strong associations with both death and ICUA, but the strength and even direction of these associations differed substantially across these outcomes. From logistic regression analysis, immunosuppression due to disease (OR 1.39, 95% CI 1.10–1.76), type-2 diabetes (OR 1.31, 95% CI 1.17–1.46), chronic respiratory disease (OR 1.19, 95% CI 1.05–1.35), age (OR 1.56 for each 10 year age increment, 95% CI 1.51–1.61), and being male (OR 1.54, 95% CI 1.42–1.68) were strongly associated with deaths due to COVID-19. The regression was adjusted for other comorbidities including type-1 diabetes, chronic liver disease, serious mental illness, chronic renal disease, chronic neurological condition, chronic heart disease, hypertension, obesity and asthma, none of which were significantly associated with death (BH test). Having any comorbidity other than these was recorded in the dataset as "other comorbidity" and appeared to be a protective factor (OR death, 0.87, 95% CI 0.80–

0.95). Some self-reported ethnicities, compared to white British, were associated with substantially increased risk of death (e.g., Indian (OR 1.84, 95% CI 1.42–2.73)) risk of death. Asymptomatic testing was associated with substantially lower risk of death (OR 0.29, 95% CI 0.18–0.45). The estimated ORs of deaths are detailed in Table 3 and illustrated in Fig A in S1 Text.

**Table 3. Estimated odd ratios (ORs) from adjusted logistic regressions and importance (*Imp*) scores of death and intensive-care unit admission (ICUA) outcomes.**

| | Death outcome | | | | ICUA outcome | | | |
|---|---|---|---|---|---|---|---|---|
| | OR | 95% CI | Pr(>\|z\|) | *Imp* | OR | 95% CI | Pr(>\|z\|) | *Imp* |
| **Comorbidities:** | | | | | | | | |
| Immunosuppr. disease | 1.392 | 1.1–1.76 | 0.006 | 0.25 | 0.826 | 0.63–1.07 | 0.157[†] | -0.13 |
| T2 diabetes | 1.307 | 1.17–1.46 | 0.000 | 0.18 | 1.018 | 0.9–1.15 | 0.778[†] | -0.05 |
| T1 diabetes | 1.228 | 0.85–1.77 | 0.275[†] | 0.07 | 0.414 | 0.27–0.63 | 0.000 | -1.09 |
| Chronic liver | 1.215 | 0.89–1.64 | 0.209[†] | 0.11 | 1.146 | 0.83–1.58 | 0.406[†] | 0.00 |
| Chronic respiratory disease | 1.188 | 1.05–1.35 | 0.008 | 0.08 | 0.830 | 0.72–0.96 | 0.011 | -0.21 |
| Obesity (clinical) | 1.163 | 1.01–1.33 | 0.030[†] | 0.08 | 3.371 | 2.9–3.92 | 0.000 | 0.87 |
| Serious mental illness | 1.087 | 0.63–1.82 | 0.755[†] | 0.04 | 2.575 | 1.5–4.46 | 0.001 | 0.49 |
| Chronic renal disease | 1.081 | 0.94–1.25 | 0.284[†] | 0.15 | 0.672 | 0.57–0.79 | 0.000 | -0.21 |
| Chronic neurological cond. | 1.064 | 0.93–1.22 | 0.381[†] | 0.08 | 0.322 | 0.27–0.39 | 0.000 | -1.01 |
| Chronic heart disease | 1.017 | 0.91–1.14 | 0.770[†] | 0.05 | 0.481 | 0.42–0.55 | 0.000 | -0.45 |
| Hypertension | 1.003 | 0.91–1.1 | 0.958[†] | 0.00 | 1.578 | 1.42–1.76 | 0.000 | 0.30 |
| Other comorbidity | 0.871 | 0.8–0.95 | 0.003 | -0.05 | 1.314 | 1.19–1.45 | 0.000 | 0.21 |
| Asthma | 0.869 | 0.75–1.01 | 0.070[†] | -0.10 | 1.512 | 1.29–1.77 | 0.000 | 0.25 |
| **Ethnicities:** | | | | | | | | |
| White and Asian | 2.401 | 0.92–6.11 | 0.066 | 0.00 | 2.451 | 0.95–6.91 | 0.073[†] | 0.00 |
| Other black | 2.204 | 1.34–3.61 | 0.002 | 0.42 | 3.583 | 1.96–7.03 | 0.000 | 1.18 |
| White and black Caribbean | 1.996 | 1.03–3.83 | 0.038 | 0.00 | 2.570 | 1.23–5.84 | 0.017 | 0.76 |
| White and black African | 1.842 | 0.78–4.15 | 0.149[†] | 0.00 | 3.439 | 1.33–10.75 | 0.018 | 0.61 |
| Indian | 1.838 | 1.42–2.37 | 0.000 | 0.41 | 2.443 | 1.86–3.23 | 0.000 | 0.76 |
| Chinese | 1.784 | 0.85–3.71 | 0.122[†] | 0.00 | 10.224 | 3.92–35.06 | 0.000 | 1.85 |
| Pakistani | 1.709 | 1.28–2.28 | 0.000 | 0.33 | 1.158 | 0.87–1.54 | 0.314[†] | 0.04 |
| Other mixed | 1.584 | 0.78–3.07 | 0.187[†] | 0.00 | 3.069 | 1.5–6.81 | 0.003 | 1.03 |
| Black Caribbean | 1.499 | 1.02–2.2 | 0.040 | 0.29 | 5.247 | 3.26–8.84 | 0.000 | 1.60 |
| Bangladeshi | 1.376 | 0.7–2.61 | 0.338[†] | 0.00 | 3.086 | 1.6–6.32 | 0.001 | 1.11 |
| Other Asian | 1.265 | 0.97–1.65 | 0.084 | 0.19 | 3.183 | 2.41–4.25 | 0.000 | 1.01 |
| Other white | 1.076 | 0.83–1.39 | 0.585[†] | 0.03 | 2.721 | 2.09–3.57 | 0.000 | 1.01 |
| Eth. unrecorded | 0.969 | 0.86–1.09 | 0.607[†] | -0.05 | 0.160 | 0.13–0.19 | 0.000 | -1.78 |
| Other eth. | 0.966 | 0.72–1.28 | 0.809[†] | 0.03 | 3.711 | 2.75–5.08 | 0.000 | 1.03 |
| Black African | 0.922 | 0.62–1.33 | 0.672[†] | -0.04 | 4.170 | 2.78–6.46 | 0.000 | 1.35 |
| Eth. unknown | 0.859 | 0.75–0.99 | 0.032[†] | -0.10 | 0.770 | 0.67–0.88 | 0.000 | -0.26 |
| White Irish | 0.493 | 0.25–0.92 | 0.032[†] | -0.19 | 0.933 | 0.51–1.68 | 0.818[†] | 0.00 |
| **Other:** | | | | | | | | |
| Sex unknown | 1.900 | 0.76–4.74 | 0.165[†] | 0.00 | 0.395 | 0.08–1.45 | 0.205[†] | 0.00 |
| Age (x10 years) | 1.560 | 1.51–1.61 | 0.000 | 0.02 | 0.764 | 0.74–0.78 | 0.000 | -0.02 |
| Sex male | 1.543 | 1.42–1.68 | 0.000 | 0.13 | 1.735 | 1.59–1.89 | 0.000 | 0.16 |
| Immunosuppr. treatment | 1.229 | 0.98–1.54 | 0.072[†] | 0.10 | 1.793 | 1.41–2.28 | 0.000 | 0.55 |
| Admission day | 0.795 | 0.76–0.83 | 0.000 | -0.24 | 0.666 | 0.64–0.7 | 0.000 | -0.39 |
| Pregnancy | 0.714 | 0.3–1.52 | 0.414[†] | 0.00 | 0.339 | 0.2–0.57 | 0.000 | -0.19 |
| Asymptomatic testing | 0.291 | 0.18–0.45 | 0.000 | -0.82 | 0.517 | 0.35–0.74 | 0.000 | -0.44 |

P values that do not test significant according to the Benjamini-Hochberg procedure are marked with a dagger(†).

Among co-morbidities, obesity (OR 3.37, 95% CI 2.90–3.29), serious mental illness (OR 2.57, 95% CI 1.51–4.46), hypertension (OR 1.58, 95% CI 1.42–1.76), asthma (OR 1.51, 95% CI 1.29–1.77), and "other comorbidity" (OR 1.31, 95% CI 1.19–1.45) were strongly positively associated with ICU admission (Fig B in S1 Text and Table 3). Each of these had far weaker or even negative associations with death. Some features associated with increased risk of death such as chronic respiratory disease were negatively associated with ICUA (OR 0.83, 95% CI 0.72–0.96). No ethnicity was negatively associated with ICUA compared to white British although there was substantial variation across these. Other factors associated with ICUA included immunosuppression due to treatment (OR 1.79, 95% CI 1.41–2.28) and male sex (OR 1.73, 95% CI 1.58–1.89). Old age (OR 0.76, 95% CI 0.74–0.78 for each 10-year increment), asymptomatic testing (OR 0.52, 95% CI 0.35–0.74), and pregnancy (OR 0.34, 95% CI 0.20–0.57) were associated with decreased ICUA. The associations of each predictor with death and with ICUA are illustrated in Fig 2, highlighting some contrasts in direction and magnitude while other risk factors appear more consistently associated with the two outcomes. The overall associations obtained from the GBDT model were consistent with the logistic model results.

The receiver operating characteristic (ROC) curves for the logistic regression models are plotted in Fig 3. The ROC-Area Under the Curve (AUC) scores for the logistic regression classifiers were 0.68 (95% CI 0.65–0.71) and 0.80 (95% CI 0.78–0.82) for death and ICUA outcome predictions, respectively. Generalized collinearity diagnostics by means of variance inflation factor (VIF) excluded severe collinearity (VIFs <2, Table 4, see also reference [21]). The scores of GBDT for classification task were 0.68 (95% CI 0.66–0.71) and 0.83 (95% CI 0.82–0.85) for the death and ICUA outcome predictions, respectively. In addition to outcome prediction, the GBDT analysis with Shapley value explanations yielded the impact of each feature on both death and ICUA outcome for each single patient (summarised in Figs C and D in S1 Text).

We contrasted the Shapley values for impacts on death and ICUA in Fig 4. All patients with obesity, serious mental illness, immunosuppressing treatment, male sex, asymptomatic admission, and those whose self-reported ethnicity was other black, Indian, black Caribbean, other Asian, other white, and NA had concordant impacts to death and ICU admission. In almost all asthma patients, it is possible to appreciate negative impact on death and positive impact on ICUA. Patients with type-1 diabetes, chronic renal disease, or chronic neurological disease show positive association with death and negative association with ICUA outcome, although with very dispersed Shapley value distributions. Upon visual inspection, the scatter points for chronic liver disease, type-1 diabetes, chronic neurological, and chronic heart comorbidities show two (or more) clusters with respect to the impact on death. The hypertension scatter plot displays a neat partition with respect to the impact on ICUA outcome, showing that this variable was associated with ICUA. Its impact on death is less clear, with patients having discordant or concordant Shapley values for death. The cases of type-2 diabetes and chronic respiratory disease appear diametrically opposite to these, as all patients with such conditions had positive Shapley values for death with qualitatively different impacts on ICU outcome.

Stratifying on ICUA yields marginally higher ROC-AUC scores (logistic regression 0.69 (95% CI 0.66–0.72), GBDT 0.70 (95% CI 0.67–0.72) compared to death prediction obtained without ICUA prediction. In fact, ICUA is a very strong predictor of death (OR 2.25, 95% CI 2.04–2.48) but is markedly correlated to other features (Fig 1). The full results are summarised in Figs E-F and Table A in S1 Text.

The features were ranked according to their median ORs and their importance scores *Imp* (defined in Materials and methods), showing that these two are ordinally associated in both death (Spearman's $\rho = 0.47$, P = 0.005) and ICUA outcomes (Spearman's $\rho = 0.97$, P = $13 \times 10^{-22}$), as shown in Fig 5. The explanation model for the GBDT was therefore largely

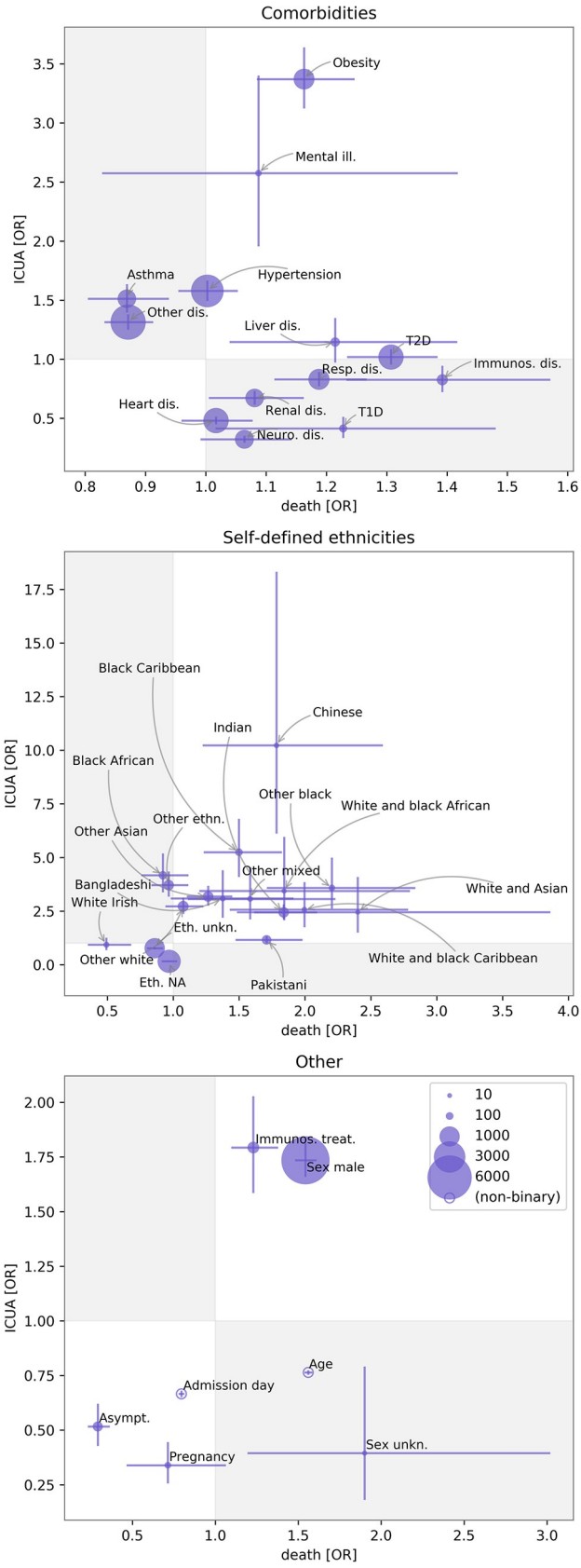

**Fig 2. Contrasting odd ratios (ORs) of death with ORs of intensive care unit admission (ICUA).** Features are grouped into comorbidities, self-defined ethnicities, and others (top to bottom). For binary variables, marker sizes are proportional to the frequencies of the exposure. Error bars are 68% confidence intervals (CIs). Gray and white regions correspond to discordant and concordant associations. The figure highlights mismatches in the ORs of a number of variables, e.g., asthma and "other comorbidity" were risk factors for ICUA but protective for death outcome. Chronic respiratory disease was a risk factor for death but negatively associated with ICU admission. For most ethnicities the ORs of death and ICUA were concordant in sign but of different magnitude. Abbreviations: Mental ill.: serious mental illness; Resp. dis.: respiratory disease; Neuro. dis.: neurological disease; Immunos. dis.: immunosuppression due to disease; T2D: type-1 diabetes; T1D: type-2 diabetes; Eth. NA: ethnicity unrecorded; Immunos. treat.: immunosuppression due to treatment; Asymp.: asymptomatic–meaning that testing was not due to the presence of COVID-19 symptoms.

consistent with the interpretable logistic linear model. The analysis of SHAP main effect also revealed the *non-linear* relations between outcomes and the age and admission day (Figs 6 and 7). The probability of death rose above 30 years of age. Likelihood of ICU admission decreased markedly above 60.

## Discussion

This cohort study investigated the association between patient characteristics (demographics and comorbidities) and severe outcomes with COVID-19 using a large national dataset in England (the CHESS database). Our findings on many factors were largely consistent with the patterns observed worldwide in studies on patients infected with SARS-CoV-2 [22–34]. Both logistic and GBDT models predicted admission to ICU more accurately than death.

Obese patients were approximately 3.4-fold more likely to be admitted to ICU (the strongest association for any co-morbid condition), while the association with mortality was small and non-significant (OR 1.16, BH test). In a US study involving 3615 patients, patients with a body mass index (BMI) between 30 and 35 were 2-fold more likely to reach the ICU and those with a BMI of over 35 were 3-fold more likely, when compared to BMIs of less than 30 [22]. These very high levels of ICUA in our and other works, as well as the contrastingly weaker association with COVID-19 mortality, could be explained by clinicians tending to, relatively, over-admit obese patients to ICU. It could reflect ICUA being very effective in reducing

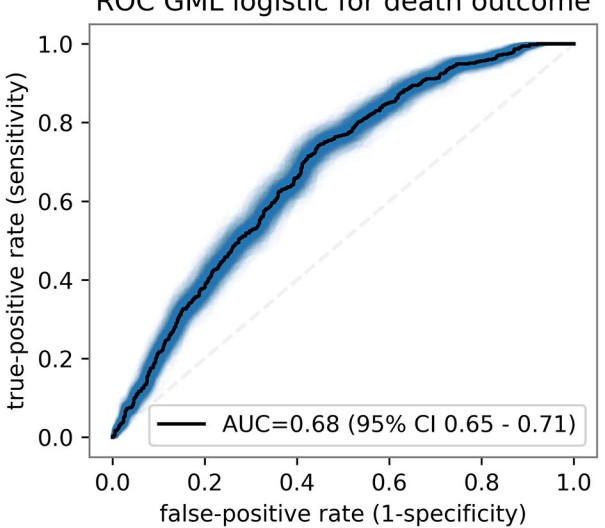
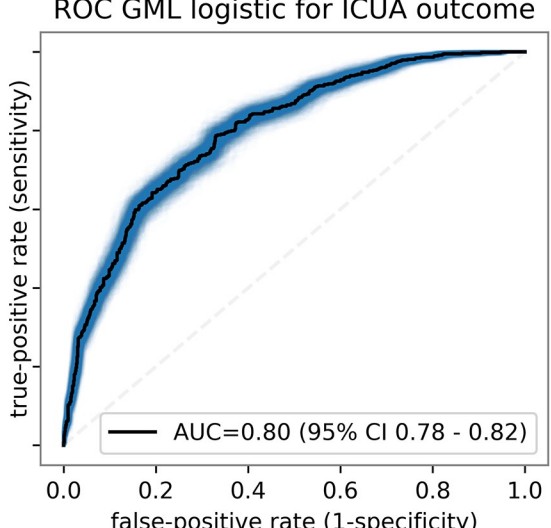

**Fig 3. ROC curves (C-statistics) of the logistic regression classifiers over the validation set.** Confidence intervals are obtained by means of bootstrapping.

**Table 4. Variance inflation factors (VIFs) for the logistic regressions.** VIF scores are always smaller than two, excluding serious collinearity issues.

|  | VIF death | VIF ICUA |
|---|---|---|
| Age (x10 years) | 1.29 | 1.19 |
| Hypertension | 1.26 | 1.28 |
| Chronic heart disease | 1.25 | 1.21 |
| T2 diabetes | 1.19 | 1.18 |
| Other comorbidity | 1.18 | 1.18 |
| Chronic renal disease | 1.15 | 1.14 |
| Obesity (clinical) | 1.13 | 1.07 |
| Eth. NA | 1.12 | 1.07 |
| Chronic respiratory disease | 1.10 | 1.10 |
| Chronic neurological cond. | 1.08 | 1.04 |
| Admission day | 1.07 | 1.13 |
| Eth. unknown | 1.07 | 1.07 |
| Immunosuppr. treatment | 1.06 | 1.06 |
| Immunosuppr. disease | 1.05 | 1.05 |
| Asthma | 1.05 | 1.05 |
| Other Asian | 1.05 | 1.03 |
| Sex male | 1.05 | 1.03 |
| Indian | 1.04 | 1.03 |
| Pakistani | 1.04 | 1.04 |
| Asymptomatic testing | 1.04 | 1.09 |
| Other ethn. | 1.04 | 1.02 |
| Other white | 1.03 | 1.02 |
| Black African | 1.03 | 1.02 |
| Other black | 1.02 | 1.01 |
| Chronic liver | 1.02 | 1.02 |
| Black Caribbean | 1.02 | 1.01 |
| T1 diabetes | 1.02 | 1.02 |
| Serious mental illness | 1.01 | 1.02 |
| Other mixed | 1.01 | 1.01 |
| White and black Caribbean | 1.01 | 1.01 |
| Sex unknown | 1.01 | 1.01 |
| Bangladeshi | 1.01 | 1.01 |
| Pregnancy | 1.01 | 1.03 |
| White and Asian | 1.01 | 1.01 |
| White and black African | 1.01 | 1.00 |
| Chinese | 1.01 | 1.00 |

mortality in this group and is an important area for further research [35]. Hypertension, and asthma were associated with ICU admission but not death. Others have reported increased risk of severe COVID-19 among asthmatics, with the increase driven only by patients with non-allergic asthma [27]. Hypertension has been associated with severe COVID-19 disease in previous univariable studies but there is no clear evidence that hypertension is an independent risk factor [29].

Black or Asian minority ethnic groups showed higher odds of death and substantially higher odds of ICU admission in our data compared to white British patients. Similar findings to ours have been demonstrated UK-wide. Multivariable analyses from large multi-ethnic

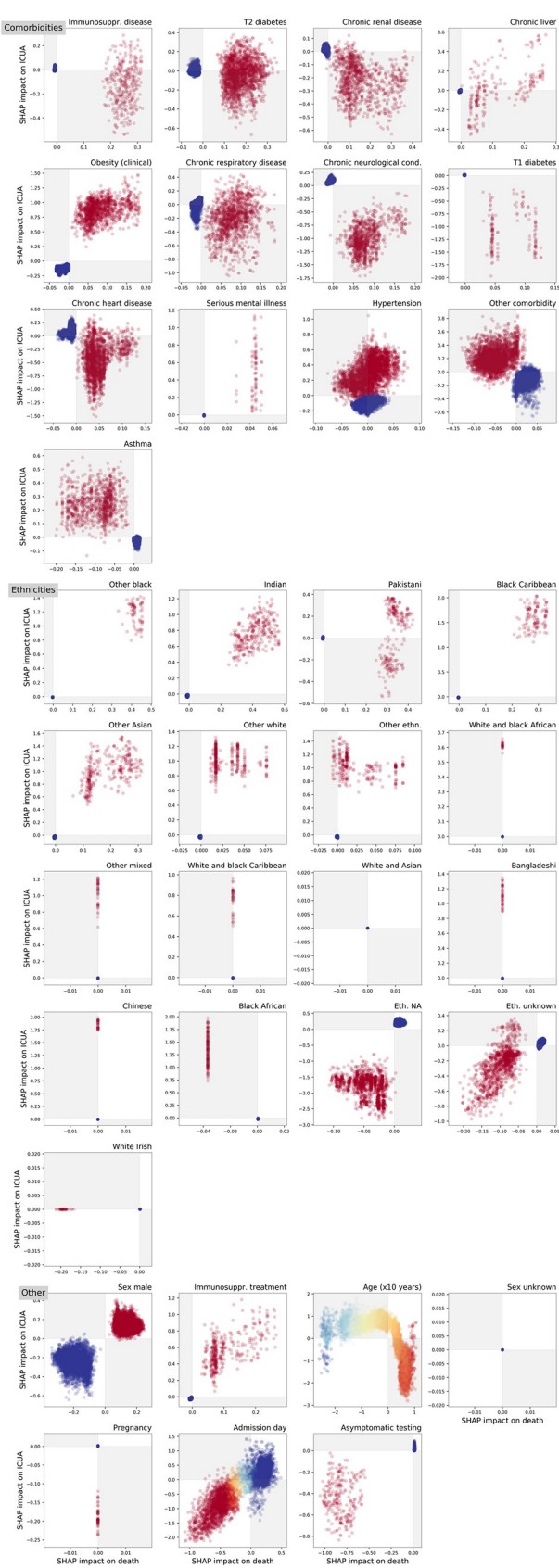

**Fig 4. Contrasting Shapley values for impact on death and intensive-care unit admission ICUA for all variables included in this study.** Each marker in the scatter plots corresponds to an in-patient. Colours from red to blue indicate the value of the underlying variable (in binary variables, red colour means feature is present, blue otherwise; in age feature, red to blue shades correspond to old to young ages; in admission day, red to blue shades correspond to early to late dates). The explanation models assigned a concordant (discordant) impact on death and ICUA to the patients in the white (grey) regions. The scatter plots expose not only the importance of a potential risk factor but also its range of effects over the cohort. All patients with immunosuppression disease, type-2 diabetes, liver and respiratory disease, and Pakistani self-defined ethnicity had positive Shapley values from death, with impact on ICU ranging from negative to positive values, thus suggesting that these conditions were always leaning towards death but sometimes not consistently towards ICUA. Conversely hypertension always have positive impact on ICUA whilst can either have positive or negative impact on death for different patients. The Shapley values for death for many features appear clustered (T1 diabetes, chronic liver, neurological, and hearth disease comorbidities), thus suggesting the presence of different groups under the same labels with different effect on patient health. Inspection of the age pattern suggests the presence of a group of young patients (blue markers) with negative impact on both age and ICUA outcome, old-age patients with positive impact on death and negative impact on ICUA outcome, and intermediate-age patients with impacts negative on death and positive on ICUA outcome. For abbreviations, see the caption of Fig 2.

cohorts have suggested that Asian and black patients group experienced an excessive level of mortality, hospital admission, and intensive care admission even when differences in age, sex, deprivation, geographical region, and some key comorbidities were taken into account [5,28,30,31]. White Irish ethnicity was non-significantly associated with lower risk of death (OR 0.49, BH test). This finding, adjusted for all covariates, echoes findings in an earlier study comparing death rates standardised for age and region using census data [30]. Chinese ethnicity predicted ICU admission (OR 10.22 with respect to the white British baseline) most strongly, followed by black Caribbean (OR 5.25). For these and other minority groups the association with ICU admission far exceeded that of death. An unrecorded or unknown ethnicity was strongly negatively associated with ICU admission, but not strongly associated with death. This may indicate increased recording of ethnicity on ICU admission, a potential cause of bias in estimating true differences in risk of ICU admission across ethnicities.

Age, type-1 diabetes, and neurological, heart, and respiratory diseases were negatively associated with ICU admission but not death. Age and chronic respiratory disease were strongly positively associated with death. Data gathered across the USA showed that deaths are 90 times higher in the 65–74 age group than the 18–29 age group and 630 times higher in the 85 and older group [36]. This may reflect judgements of limited capacity to benefit from ICU admission due to age and some co-morbidities. Type-2 diabetes is broadly reported to be associated with poor outcome in COVID-19 patients, while studies reporting outcome for type-1 diabetes are rare [32,33]. A national general practice based analysis in England demonstrated that both type-1 and type-2 diabetes are associated with increased risk of in-hospital death with COVID-19 [34]. Our multiply adjusted analysis of the CHESS dataset confirmed that type-2 diabetes had a strong association with mortality (and non-significant association with ICU admission), while type-1 diabetes' association was positive but not statistically significant. On the other hand, type-1 diabetes was negatively associated with ICUA outcome. There is uncertainty regarding the effect of diabetes and glycaemic control on COVID-19 outcome. Whilst some suggest a 3-fold increase in intensive care admission and death [24], others found no association between glycaemic control and severe outcome [26]. Potential mechanisms for effects could include hyperinsulinemia or the interaction of SARS-CoV-2 with ACE2 receptors expressed in pancreatic β cells [32,37].

Male sex was positively and similarly associated with both ICU admission and death. The increased risk of male deaths is consistent with worldwide data, in which, on average, 1.4-fold more men than women have died from SARS-CoV2, with some countries reporting greater than 2-fold male deaths [38]. Increased expression of the ACE2 receptor may occur in men and has been suggested as a possible explanation for this finding [25]. Asymptomatic testing

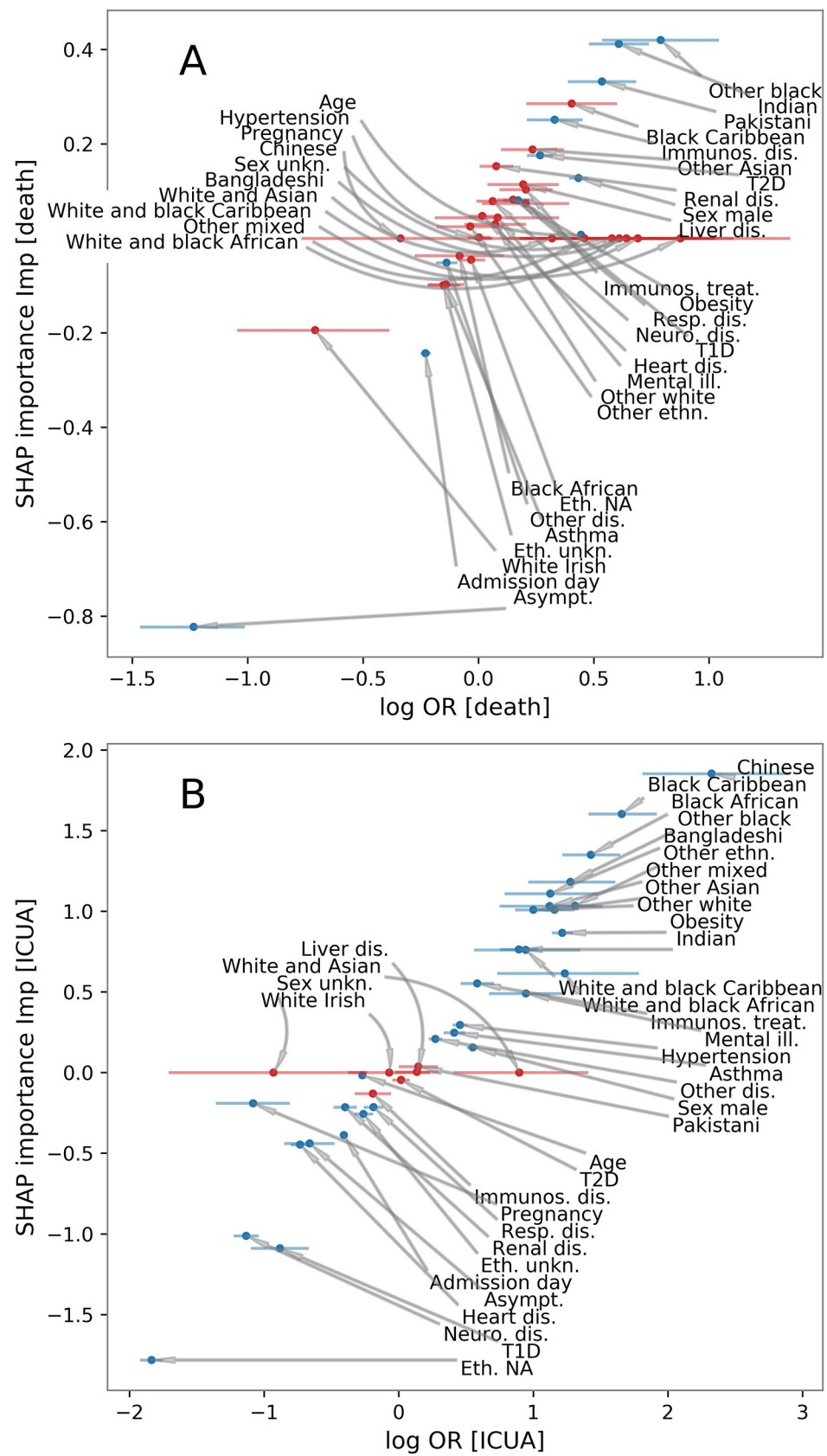

**Fig 5.** SHAP importance scores from the explanation model for GBDT vs logarithm of odd-rations (ORs) from logistic regression for death (A) and intensive-care unit (ICU) admission (B). Each point represents a feature (see Table 3). Red markers correspond to the features whose association with the outcome was not significant according to the logistic regression. The x-axis errorbars comprise 68% confidence intervals. The SHAP importance *Imp* allows us to assess to what extent a feature contributes to the GBDT prediction. This plot shows that these are consistent with the well-known logistic regression coefficients, despite the underlying models used to generate these two quantities are fundamentally different.

and pregnancy demonstrated a strong negative association with both death and ICU admission. These results were expected in view of NHS trusts undertaking surveillance swabs for asymptomatic people, including among elective hospital admissions.

Different machine-learning models have been leveraged to predict COVID-19 patients at risk of sudden deterioration. A study over 162 infected patients in Israel demonstrated that artificial intelligence may allow accurate risk prediction for COVID-19 patients using three models (neural networks, random trees, and random forests) [39]; a random forest model was

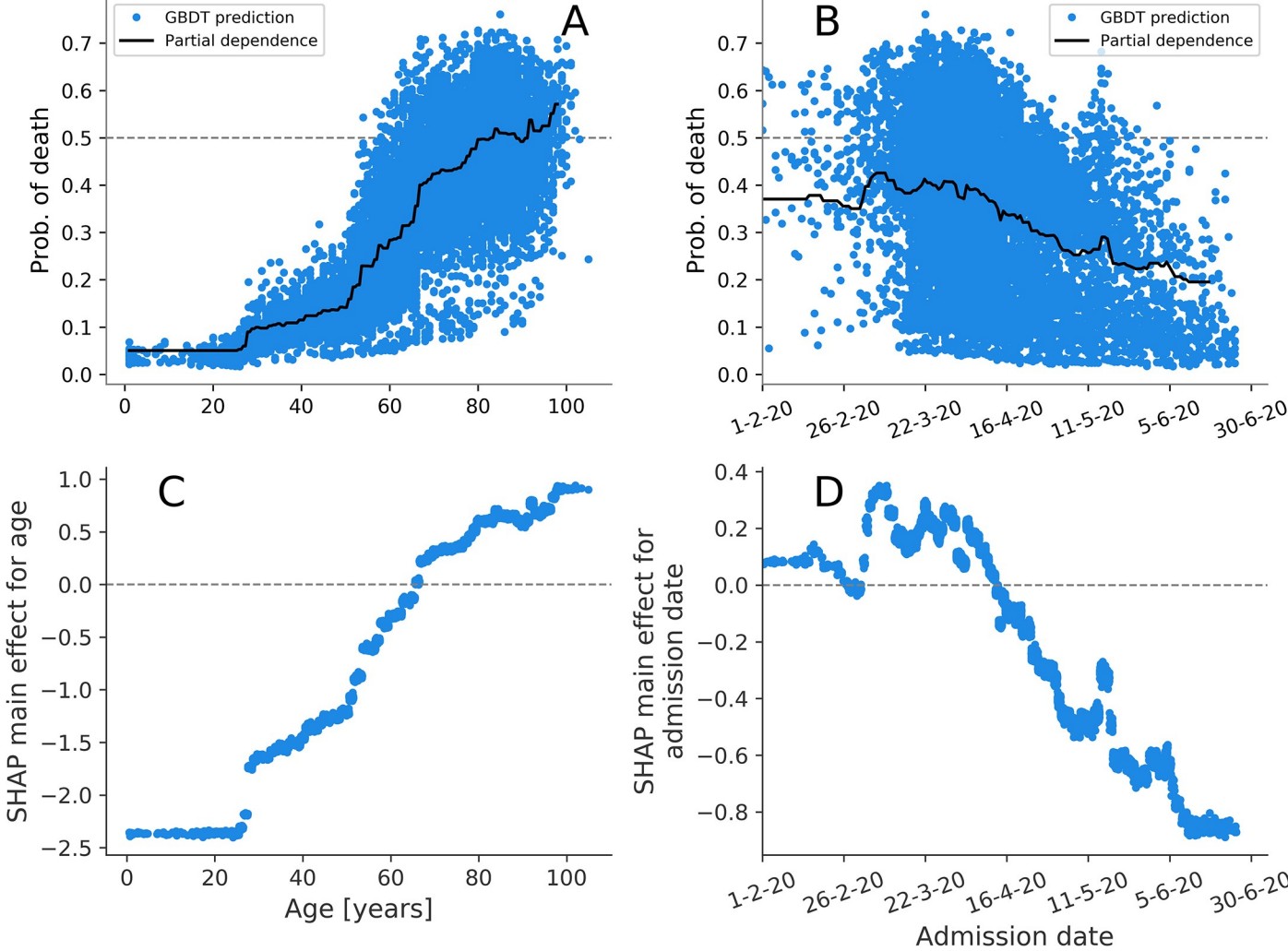

**Fig 6.** A-B) Partial dependence plots (PDPs) and probability of death predicted by GBDT for each patient in training set. C-D) SHAP main effect for age and admission date. These effects can be ascribed to the age/admission date alone, regardless of their covariates. The strong pattern in the main effect for admission date highlights the importance of incorporating timing in predictive models.

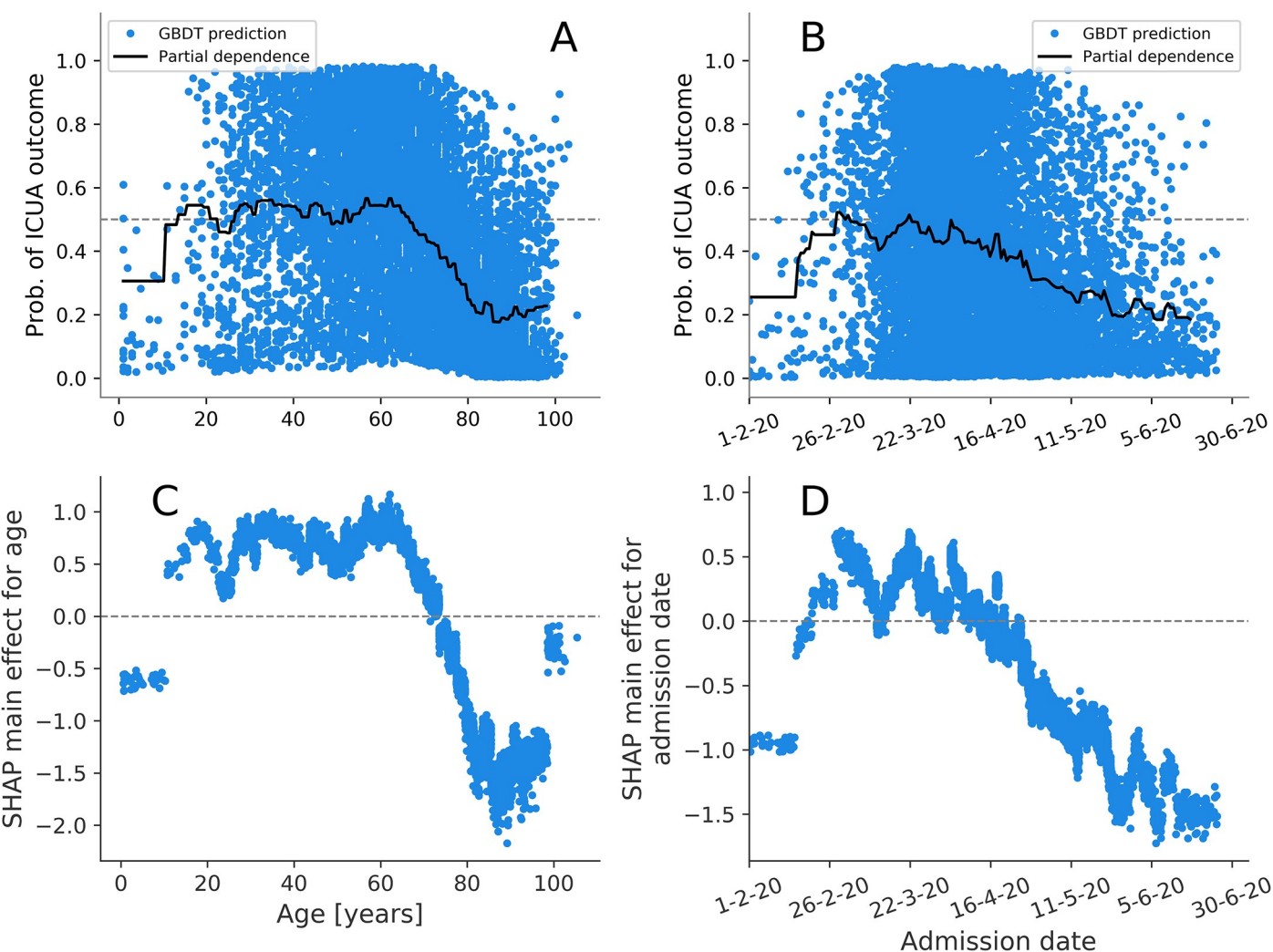

**Fig 7.** A-B) Partial dependence plots (PDPs) and probability of intensive-care unit admission predicted by GBDT for each patient in training set. C-D) SHAP main effect for age and admission date.

used over 1987 patients for early prediction of ICU transfer [40]; the GBDT model was deployed on blood-sample data from 485 patients in Wuhan, China [41]; GBDT models outperformed conventional early-warning scoring systems for ventilation requirement prediction over 197 patients [42]; deep learning and ensemble models were reported to perform well for early warning and triaging in China [9,43]. These models are very complex, but evidence indicates that mortality predictions can be obtained from more parsimonious models, upon selecting the most important features, thus facilitating more efficient implementation of machine-learning in clinical environments [44]. Despite these successes, prediction models have been found overall to be poorly reported and at high risk of bias in a systematic review [45]. A comprehensive list of relevant works is out of the scope of this paper, but it is worth underlining that machine-learning methods typically excel in outcome prediction but lack ease of interpretation of the result. In this study, we bridged the gap between performance and interpretability in machine learning for poor outcome predictions in COVID-19 patients. We trained GBDT models (see Materials and methods section) and extracted not-only their predictions, but also the extent to which each potential risk factor contributed to the prediction overall (thus

permitting comparisons with the more easily-interpretable logistic regression model) and for each patient. So-called "Shapley values" quantify such information, as summarised in Fig C and D of S1 Text for death and ICUA, respectively.

Overall, the association of patient features with the final outcome (measured by the SHAP importance scores *Imp*, see Materials and methods and Fig 5) is consistent with the logistic regression results, although the two models are intrinsically different. Moreover, for each feature, we derived an individual Shapley value for each patient, allowing us to consider the variation in effects among patients. As a first example, we discuss interpretation of type-2 diabetes. In the summary plots of Fig 4 and Figs C and D in S1 Text, the red markers correspond to type-2 diabetes patients and blue to patients without type-2 diabetes. In the summary plot for death outcome (Fig C in S1 Text, see also Fig 4), the red and blue markers are grouped into two distinct clusters. All the type-2 diabetes patients had positive Shapley values, thus showing that such a comorbidity was always associated with death, while all the other patients had nearly zero Shapley values. Conversely, in the summary plot for ICUA outcome (Fig D in S1 Text, see also Fig 4), the red markers appear scattered. Some T2-diabetes patients had positive Shapley values (positive association with ICU admission) while others had negative values (a negative association with ICU admission). The summary plots thus show not only the overall importance of a potential risk factor, but also its range of effects over the patients. In this case our interpretation is that although consistently increasing the risk of death, the presence of type 2 diabetes had more variable impact on decision making around ICU admission, in some cases apparently adding to the case for admission and in some cases diminishing it.

Being male was positively associated with both death and ICU admission. Its impacts were concordant in sign and confined within a narrow range of values. Conversely, for example, chronic renal disease and immunosuppressive treatment had low impact on predicting death for some patients, but very high impact for others, perhaps reflecting that these categories comprise a number of diverse conditions and therapies. Considering ethnicity, most minority groups were consistently and positively associated with ICUA but the impact attributed to Pakistani ethnicity were much more variable.

Shapley value analysis of the GBDT model also excels in explaining the nonlinear relations between covariates and their importance to outcome prediction. In Fig 6A, the predicted probability of death is shown to increase with age, in part due to increasing presence of comorbidities which are correlated with increasing age (Fig 1). In fact, the isolated effect of age (the SHAP main effects for age), illustrated in Fig 6C, shows a sharp rise from age 30 even if it is stripped from the interactions with the other factors. For ICUA, the SHAP main effect for age abruptly drops and even reverses from the 60th year of age (Fig 7). The abruptness may suggest an age threshold is being applied in clinical decision making on ICU admission.

During the first peak of COVID-19 epidemic healthcare services were under variable strain, with clinical expertise growing over time. Declining in-hospital mortality was observed in Italy [46] and England [47] during the first pandemic peak. This may reflect a mix of changing pressure, developing clinical expertise and variable follow-up time following admission. We included the patient's admission day in our models to allow for these effects in adjustment (logistic regression) and attribution of impact (machine learning). Hospital admission later than March decreased both death and ICUA. These results mirror the PDPs outlined in Figs 6A and 6B and 7A and 7B, showing that a local explanation technique such as the Shapley value analysis supersedes and is consistent with the global explanation of the PDPs. The performance gains of the GBDTs here are small, in part due to the fact that all but two predictors (age and admission date) are binary. Indeed, the logistic model predictions depend on a linear combination of the predictor values, which is adequate if all the predictors are binary and the classes are linearly separable. The similarity in the predictive power for these specific cases

should not shadow the other advantages of the GBDTs (including their greater generality and their ability of detecting non-linearity and variation in predictive effect).

While all our models had excellent performances, it is worth noting that prediction of ICUA outcome was significantly better than death alone prediction for both. Including laboratory test results in the predictor variable may improve death prediction [48].

In conclusion, this study confirms that, in hospitalised patients, the risk of severe COVID-19, defined as either death or transfer to intensive care unit, is strongly associated with known demographic factors and comorbidities. We found that the association of these variables with death was often qualitatively and quantitatively different from their association with ICU admission. This was consistently derived by means of two different predictive models, i.e., the standard logistic and the GBDT machine-learning models. The Shapley value explanation of the latter model also highlights the sometimes variable impact of each factor for each patient. These results allow an insight into the variable impact of individual risk factors on clinical decision support systems. We suggest that these should not only grant the optimal average prediction, but also provide interpretable outputs for validation by domain experts. Shapley values may also support analytical approaches to address the problem of characterising the group of patients for whom a prediction is incorrect. This is an important additional potential area for research and application. Shapley-value analyses allow clinical interpretation of the results from a complex machine-learning model such as the GBDT. Using these we have derived importance scores which are consistent with the better known ORs as an overall assessment of an average effect but can additionally display the extent to which this average effect is consistent across patients or highly variable among different patient groups. We recommend the wider adoption of Shapley-value analyses to support interpretation of ML outputs in clinical decision making given this capacity to communicate the variation in the effects of predictive variables. These aspects are particularly valuable to tackle COVID-19, a complex disease that can cause a variety of symptoms and clinical outcomes, depending on the patients' conditions, and rapidly overwhelm healthcare systems, thus requiring large-scale automated decision systems.

## Supporting information

**S1 Text.** Supporting Information, including: Figs A, B, C, D, E, and F, and Table A. (PDF)

## Author Contributions

**Conceptualization:** Massimo Cavallaro, Matt J. Keeling, Noel D. McCarthy.

**Data curation:** Massimo Cavallaro, Matt J. Keeling.

**Formal analysis:** Massimo Cavallaro.

**Funding acquisition:** Matt J. Keeling, Noel D. McCarthy.

**Investigation:** Massimo Cavallaro.

**Methodology:** Massimo Cavallaro.

**Software:** Massimo Cavallaro.

**Visualization:** Massimo Cavallaro.

**Writing – original draft:** Massimo Cavallaro, Haseeb Moiz, Noel D. McCarthy.

**Writing – review & editing:** Massimo Cavallaro, Haseeb Moiz, Matt J. Keeling, Noel D. McCarthy.

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
