## [Decision Letter · Decision Letter 0]

1 Apr 2021

Dear Dr. Cavallaro,

Thank you very much for submitting your manuscript "Contrasting factors associated with COVID-19-related ICU and death outcomes by means of Shapley values" for consideration at PLOS Computational Biology.

As with all papers reviewed by the journal, your manuscript was reviewed by members of the editorial board and by several independent reviewers. In light of the reviews (below this email), we would like to invite the resubmission of a significantly-revised version that takes into account the reviewers' comments.

We cannot make any decision about publication until we have seen the revised manuscript and your response to the reviewers' comments. Your revised manuscript is also likely to be sent to reviewers for further evaluation.

Sincerely,

Benjamin Muir Althouse

Associate Editor

PLOS Computational Biology

Virginia Pitzer

Deputy Editor-in-Chief

PLOS Computational Biology

Reviewer's Responses to Questions

**Comments to the Authors:**

Reviewer #1: This is a (very) large retrospective examination of the ability of different modeling techniques to predict high-risk COVID patients (patients who died and those who were admitted to the ICU) and to quantify the predictive strength of included predictors. Through their comparison of modeling techniques (logistic vs gradient-boosted tree), they were able to show the difference in predictive ability of machine learning techniques. Further, by examining both the resulting odds ratios and Shapley additive explanations (a novel approach to tool interpretability) they were able to show the relative predictive strength of included factors were similar regardless of model type. This analysis is both topically and methodologically relevant and is a great addition to the existing COVID prediction and overall prediction literature. However, there are some concerns that need to be addressed prior to publication.

Overall:

1. The study population was limited to patients who required hospitalizations (stated in the methods). This restriction in the population is not clear in other parts of the analysis (though it is stated in the conclusion) but has a very meaning impact on the generalizability of the study results.

2. It is not clear to me if the main purpose of the paper is to compare the predictive modeling approaches and how predictors ranked by model or if it was the comparison of ICU vs Death outcome predictions tools or if predictor importance was the primary goal. I think the abstract and introduction would benefit from a hypothesis type sentence outlining the primary and secondary aims of the work.

3. What is the vision for the use of the Shapley values or ORs in the clinical decision-making space?

Methods

1. Why was a gradient-boosted tree selected for over an alternative ML approach (random forest, neural net, etc)?

2. It is great that authors examined the VIF to ensure that collinearity was not a concern during their interpretation of predictor importance -nice job. However, it is not clear to me that the authors accounted for the large number of pvalues they examined. Was any multiple-testing approach applied to the results, if so, which one and why was that the approach selected?

3. How were comorbidities selected for and what justified specific comorbidities to be examined individually vs. grouped?

4. The employed variable importance approach for the GBDT (Shapley Additive Explanation) is novel and its utility is well outlined. Where other types of variable importance estimation tools (such as partial dependence plots – PDPs) considered in this analysis and why was the Shapley method selected in the end?

Results

1. Given the differences in the predictive direction of some of the included features between death and ICU admission, it would be very helpful to have a breakdown of how these to outcomes overlapped – how many patients died within the ICU and what not. Further, if you stratify on ICU admission within the death prediction, is there a difference in predictive validity?

2. I am not sure what this sentence means “Generalized collinearity diagnostics by means of variance inflation factor (VIF) excluded severe collinearity (Table 4).” Based on the table heading, it seems like no VIFs were > 2 but that is not clear in the written results.

3. For comparing the ORs and Shapley Values, it would be helpful to see of the predictor order differs in a single plot. This is somewhat displayed by Figure 5 but I still found myself looking at the supplement and comparing the predictor order in my head.

4. Since the logistic and the GBDT were similarly predictive, it surprises me that the Shapley values show interaction importance and non-linear associations (something not included in the logistic). This makes me think that examining the Shapley results to this level of graduality may be a reach or that the GBDT ended up over fit despite hyper-parameter selection. Please speak to how these associations can be found meaningful yet a model that does not include them has essentially equal predictive validity.

Discussion:

1. This sentence “Chinese ethnicity predicted ICU admission (OR 10.2) most strongly, followed by black Caribbean (OR 5.2)” should incorporate that white British is the baseline. That is stated earlier but as a reader I had to go back and find the reference while ready.

2. The fact that the GBDT and the logistic regression had similar predictive validity is not discussed in the discussion. Given that they are similarly predictive, is there a reason to use the GBDT over the logistic regression in this scenario (or vice versa). The authors removed the interpretability issue of GBDT – which is great – so which should I choose?

Minor

1. Missing word or typo in the following sentence: “A GBDT aggregates a large number of weak prediction models, in this case decision trees, into a robust prediction algorithm, where the presence of many trees mitigates the errors due a single-tree prediction.”

2. A few typos found in the results – examples below:

a. Space between 95% CI in some and not other results

b. “associated with death” to “associated to death”

c. Inconsistent rounding

3. Discuss associations in the same direction (predictor to outcome) through the results. Currently, it varies from sentence to sentence and is hard to follow.

Reviewer #2: Overall, I appreciate the size of this study compared to many others in the space with almost 14,000 cases and this seems to meet an important need for addressing the causes of adverse outcome in COViD-19 with sufficient sample size. Also, the approach of feature-wise calculation of odds ratios and assessment of a subsequent model with all features is clearly laid out and easy to follow. The paper agrees with some other published associations which the authors do a good job of describing and I view that these findings and their substantial underlying dataset are important to add to demonstrate such concordance or divergence.

I find the assessment of model performance with Shapley values as well as the calculation of Variance Inflation Factors to assess collinearity to also be a valuable added point of diligence in the analysis. The conclusions drawn seem well supported by the methods used as the authors are not trying to extrapolate mechanistic or far-reaching explanations but are instead trying to clearly demonstrate the associations between various factors and outcome.

I have a few small improvements that I would suggest:

- One very small point, there seems to be two acronyms for the same entity: ICA and ICUA both for Intensive Care Unit Admission. I'm assuming these are meant to be the same thing in my review.

- Shapley value analysis is a compelling way to present and describe model behavior and the authors do a good job of discussing the implications of this. One thing that would also be interesting to show are the patients for whom the prediction of mortality or ICU admission was most incorrect and which features were present for those patients. This would be a value-added part of discussing predictive model behavior.

- Figure 2 is particularly interesting and does a good job of summarizing the various relationships between features, death, and ICU admission. The self-defined ethnicities panel in this Figure can be a bit challenging to follow since so many of the labels are located apart from the points to which they correspond although admittedly I don't see a way to improve this. Perhaps some color-coding of the various points and their CI bars would by ethnicity category would help.

- One additional small item that I think would be a nice addition is some explanation of how the demographics and deaths datasets were joined together as these cannot be provided by the authors upon publication, someone wishing to recapitulate or extend these findings may benefit from this detail.

**Have all data underlying the figures and results presented in the manuscript been provided?**

Reviewer #1: None

Reviewer #2: **No: **The authors describe that data on these cases were obtained from CHESS which is a repository of patients infected with COVID-19. Death data were obtained from Public Health England which, as authors describe, is not permissible for public disclosure This is understandable and true of many datasets so I would not hold this against the authors as they have provided the identity of the data sources so presumably, one could procure their source data from these locations.

PLOS authors have the option to publish the peer review history of their article (what does this mean?). If published, this will include your full peer review and any attached files.

Reviewer #1: No

Reviewer #2: **Yes: **Peter McCaffrey
---

## [Decision Letter · Decision Letter 1]

27 May 2021

Dear Dr. Cavallaro,

We are pleased to inform you that your manuscript 'Contrasting factors associated with COVID-19-related ICU admission and death outcomes in hospitalised patients by means of Shapley values' has been provisionally accepted for publication in PLOS Computational Biology.

Best regards,

Benjamin Muir Althouse

Associate Editor

PLOS Computational Biology

Virginia Pitzer

Deputy Editor-in-Chief

PLOS Computational Biology

Reviewer's Responses to Questions

**Comments to the Authors:**

Reviewer #1: You did an outstanding job updating the manuscript and and no additional edits or adjustments are needed. This study is incredibly interesting and valuable. Nice Job.

Reviewer #2: I appreciate the authors' openness to reviewer comments.

**Have the authors made all data and (if applicable) computational code underlying the findings in their manuscript fully available?**

Reviewer #1: **No: **Some of the data contains confidential information preventing it from public release. I find this justification to be valid and these data restrictions should not prevent this paper from being published in PLOS.

Reviewer #2: Yes

PLOS authors have the option to publish the peer review history of their article (what does this mean?). If published, this will include your full peer review and any attached files.

Reviewer #1: **Yes: **Margaret L Lind

Reviewer #2: **Yes: **Peter McCaffrey

---

## [Editor Report · Acceptance letter]

15 Jun 2021

PCOMPBIOL-D-21-00047R1 

Contrasting factors associated with COVID-19-related ICU admission and death outcomes in hospitalised patients by means of Shapley values

Dear Dr Cavallaro,

I am pleased to inform you that your manuscript has been formally accepted for publication in PLOS Computational Biology. Your manuscript is now with our production department and you will be notified of the publication date in due course.

With kind regards,

Katalin Szabo
